# DNA-damage induced cell death in *yap1;wwtr1* mutant epidermal basal cells

Jason KH Lai[1]*†, Pearlyn JY Toh[1]†, Hamizah A Cognart[1], Geetika Chouhan[2], Timothy E Saunders[1,3,4,5]*

[1]Mechanobiology Institute, National University of Singapore, Singapore, Singapore; [2]Department of Biological Sciences, Tata Institute of Fundamental Research, Mumbai, India; [3]Department of Biological Sciences, National University of Singapore, Singapore, Singapore; [4]Institute of Molecular and Cell Biology, A*Star, Singapore, Singapore; [5]Warwick Medical School, University of Warwick, Coventry, United Kingdom

**Abstract** In a previous study, it was reported that Yap1 and Wwtr1 in zebrafish regulates the morphogenesis of the posterior body and epidermal fin fold (Kimelman et al., 2017). We report here that DNA damage induces apoptosis of epidermal basal cells (EBCs) in zebrafish *yap1*[-/-]*;wwtr1*[-/-] embryos. Specifically, these mutant EBCs exhibit active Caspase-3, Caspase-8, and γH2AX, consistent with DNA damage serving as a stimulus of the extrinsic apoptotic pathway in epidermal cells. Live imaging of zebrafish epidermal cells reveals a steady growth of basal cell size in the developing embryo, but this growth is inhibited in mutant basal cells followed by apoptosis, leading to the hypothesis that factors underscoring cell size play a role in this DNA damage-induced apoptosis phenotype. We tested two of these factors using cell stretching and substrate stiffness assays, and found that HaCaT cells cultured on stiff substrates exhibit more numerous γH2AX foci compared to ones cultured on soft substrates. Thus, our experiments suggest that substrate rigidity may modulate genomic stress in epidermal cells, and that Yap1 and Wwtr1 promotes their survival.

*For correspondence:
jasonlaikh@gmail.com (JKHL);
timothy.saunders@warwick.ac.
uk (TES)

†These authors contributed
equally to this work

Competing interest: The authors declare that no competing interests exist.

## Editor's evaluation

This manuscript identifies a novel function for YAP/TAZ in epithelial cells. Mechanical stress/strain is shown to drive genomic stress and a DNA damage response in zebrafish skin, and concurrent activation of YAP/TAZ is essential to accommodate the stress/strain by allowing elastic expansion and flattening of cells as well as preventing apoptosis.

## Introduction

The Hippo signaling pathway is widely known for its role in regulating tissue and organ size (*Johnson and Halder, 2014*). More recently, the roles of its downstream effectors, Yap1 and Wwtr1 (or widely referred to as Taz), have been described in a variety of developing tissues. In the developing heart, Wwtr1 participates in cardiomyocyte decision-making to become a trabecular cell (*Lai et al., 2018*). Additionally, in the cardiovascular system, YAP1 and WWTR1 are important for sprouting angiogenesis (*Astone et al., 2018*; *Kim et al., 2017*; *Neto et al., 2018*; *Wang et al., 2017*) and vascular stability (*Nakajima et al., 2017*). Yap1 and Wwtr1 also regulate the retinal epithelium cell fate in the eye (*Miesfeld et al., 2015*), kidney branching morphogenesis (*Reginensi et al., 2015*), and fibronectin assembly for tail morphogenesis (*Kimelman et al., 2017*). In this report, we show that Yap1 and Wwtr1 are required for the survival and growth of epidermal cells during skin development.

In the zebrafish, the non-neural ectoderm, marked by *ΔNp63*, emerges from the ventral side of the gastrulating embryo, and spreads dorsally to cover the entire embryo as a sheet of cells beneath the periderm during segmentation stages (*Bakkers et al., 2002*; *Solnica-Krezel, 2005*). Residing in this layer are the epidermal basal cells (EBCs) which are responsible for epidermal cell renewal and homeostasis (*Blanpain and Fuchs, 2006*). In mice, Yap1 and Wwtr1, acting downstream of integrin signaling, were shown to modulate EBC proliferation during wound healing (*Elbediwy et al., 2016*). However, their role in early epidermal development is poorly characterized.

In addition to their role in cell proliferation, YAP1 and WWTR1 promote cell survival. When cells are challenged with DNA damaging agents including doxorubicin (*Ma et al., 2016*) and radiation (*Guillermin et al., 2021*), YAP1, as a co-transcription factor of Teads, is localized in the nucleus to drive expression of pro-survival genes. Indeed, transgenic expression of *Yap1* in murine livers upregulates the expression of *Survivin* (*Birc5*) in hepatocytes, protecting these cells from apoptosis when treated with a pro-apoptotic agent, Jo-2 (*Dong et al., 2007*). On the other hand, genetic knockdown of *Yap1* or inhibition of YAP1 nuclear localization exaggerates apoptosis in cells exposed to chemo- and radio-therapy (*Guillermin et al., 2021*; *Ma et al., 2016*). Thus, YAP1 executes its pro-survival function in the nucleus by driving the expression of target genes.

The previous work by *Kimelman et al., 2017* investigated how Yap1 and Wwtr1 participate in the development of the fin fold by regulating the expression and organization of the extracellular matrix (ECM). In the course of this previous investigation, which focused specifically on why *yap1;wwtr1* double mutants had a thinner epidermis in the ventral fin fold, the authors did not observe apoptotic epidermal cells. However, apoptosis in other parts of the embryo was not thoroughly studied, as it was outside the focus of this previous work. Here, we show that concurrent loss of *yap1* and *wwtr1* results in aberrant apoptosis of EBCs on the yolk at the 6–10 ss. However, epidermal cell proliferation is not reduced in *yap1;wwtr1* double mutant zebrafish embryos, in contrast to adult $Yap1^{-/-};Wwtr1^{-/-}$ mice. At these developmental stages, Yap1 and Wwtr1 are localized in the EBC nuclei to putatively carry out their co-transcriptional function through the Tead-binding domain (TBD) and support epidermal cell survival. This cell death phenotype in mutant embryos is executed through the extrinsic apoptotic pathway, as apoptosis assays reveal activation of Caspase-8 and Caspase-3. Further, we show that γH2AX, a DNA damage marker, is recruited in mutant EBCs. Thus, DNA damage prompted the extrinsic apoptotic pathway in these cells (*Hill et al., 1999*). Live imaging of epidermal cells found that, leading up to this apoptosis phenotype, mutant basal cell size is inhibited after the 6 ss, unlike their sibling embryo counterparts, which continue to grow. Interestingly, human keratinocytes cultured on stiffer substrates exhibit more γH2AX foci than cells cultured on softer substrates, suggesting that the mechanical environment not only modifies cell size, but also plays a role in genomic stress (*Denais et al., 2016*; *Raab et al., 2016*; *Nava et al., 2020*). Here, we use the term 'stress' to denote the pressure on genome integrity due to, for example, mechanical or radiative perturbations. Moreover, knockdown of *YAP1* or *WWTR1* in these cells further increases the number of γH2AX foci. Thus, we propose that in response to genomic stress, Yap1 and Wwtr1 have important roles in the survival of EBCs in the course of their development.

## Results

### Yap1 and Wwtr1 are localized in the nuclei of EBCs to modulate their survival

The previous work on *yap1;wwtr1* double mutant zebrafish reported no overt cell death in the tail (*Kimelman et al., 2017*). Given the important regulatory roles of YAP1 and WWTR1, we hypothesized that cell death may be increased elsewhere during early embryonic development in such embryos. As a first exploration, we focused on the lateral yolk, on which two layers of the epidermis – the periderm and basal epidermis – reside. We performed TUNEL assays on 16–18 ss embryos as by this stage the double mutants are morphologically distinct from their siblings. We found aberrant apoptosis in the $yap1^{-/-};wwtr1^{-/-}$ (henceforth referred simply as 'mutant') epidermis on the yolk, but not in the tail as reported before (*Figure 1A*). Additionally, whereas WT embryos have regular distribution of P63-positive nuclei (an EBC marker) over the yolk, mutants have a more irregular and sparser pattern (*Figure 1A*), suggesting apoptosis of mutant EBCs. As we could only detect apoptotic fragments in older mutant embryos, we investigated this phenotype in closer detail by repeating this assay in

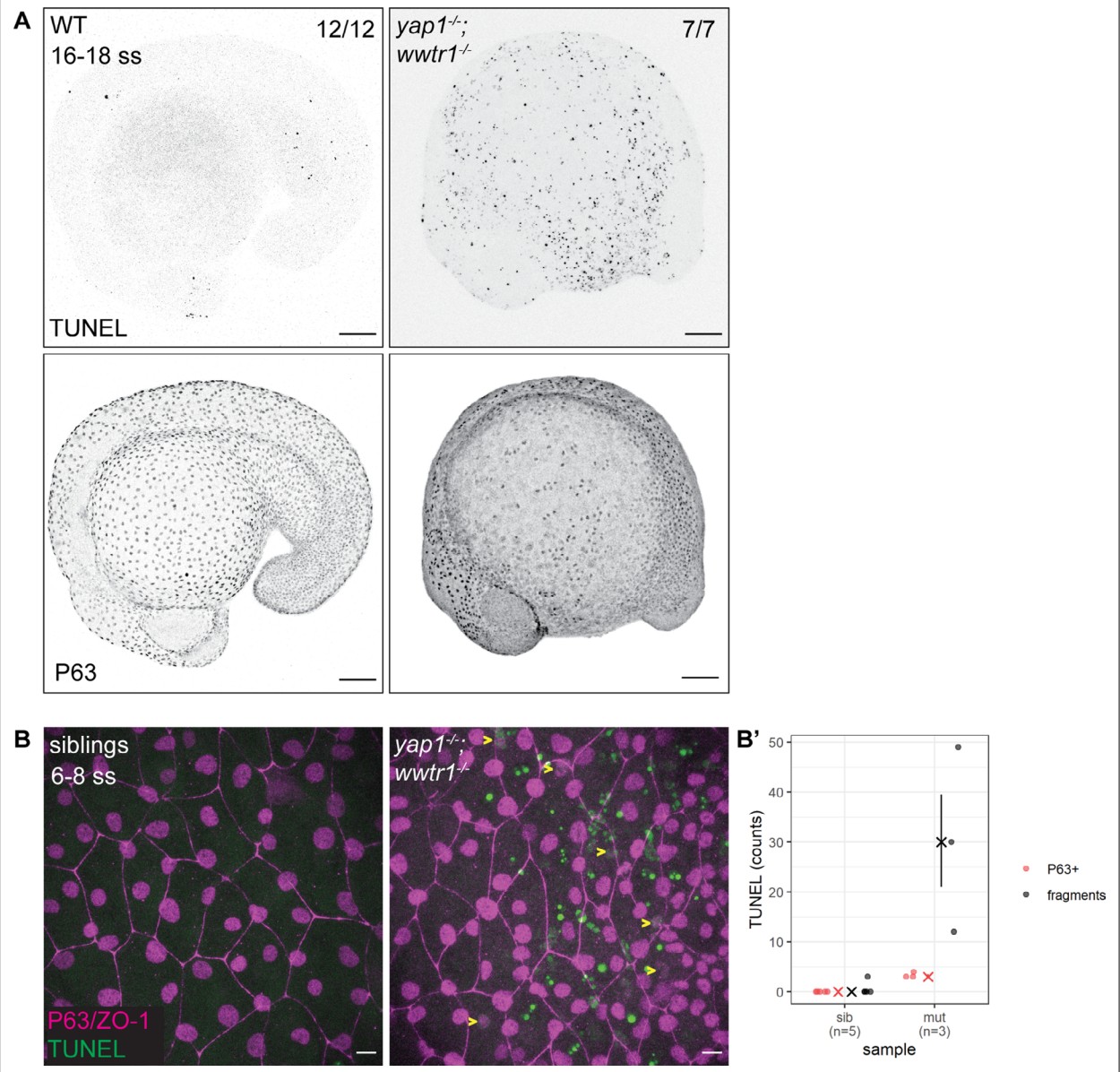

**Figure 1.** Developing zebrafish *yap1;wwtr1* double mutants show aberrant epidermal basal cell death. (**A**) Maximum intensity projections of 16–18 ss WT and mutant embryos stained with TUNEL and P63. Scale bars, 100 µm. (**B**) Maximum intensity projections of the epidermis on the lateral yolk of 6–8 ss mutant and normal siblings stained with TUNEL, P63, and ZO-1. Yellow arrowheads indicate TUNEL and P63 double positive nuclei. Scale bars, 10 µm. (**B'**) Quantification of TUNEL-positive EBC nuclei (P63+) and apoptotic fragments (fragments).

The online version of this article includes the following source data and figure supplement(s) for figure 1:

**Source data 1.** TUNEL count data for *Figure 1B'*.

**Figure supplement 1.** TUNEL analysis in early embryo.

younger embryos and imaging them with higher magnifications. Normal sibling epidermis exhibit virtually no TUNEL signal. On the other hand, mutant epidermis exhibit P63 and TUNEL co-staining (apoptotic EBCs), as well as punctate TUNEL signal (*Figure 1—figure supplement 1*). The latter corresponds to apoptotic fragments that made up the bulk of the TUNEL signal (*Figure 1B and B'*). Therefore, this phenotype suggests that Yap1 and Wwtr1 are required for EBC survival.

Using previously characterized antibodies against Yap1 and Wwtr1 (*Flinn et al., 2020*; *Kimelman et al., 2017*; *Lai et al., 2018*; *Miesfeld et al., 2015*), we investigated their cellular localization in the epidermis. Both Yap1 and Wwtr1 are localized in the nuclei of the peridermis and basal epidermis

at the tailbud stage and at the 16–20 ss (*Figure 2A*). Yap1 is also localized to the junctions of peridermal cells (*Figure 2A*; *Flinn et al., 2020*). At these developmental stages, some EBCs in the basal epidermis lose P63 expression and differentiate into ionocytes (*Jänicke et al., 2007*). By comparing P63-positive to P63-negative nuclei in the basal epidermis, we found greater expression levels of Yap1 and Wwtr1 in the P63-positive ones (*Figure 2B*), indicating Yap1 and Wwtr1 activity in EBCs.

In the nucleus, Yap1 and Wwtr1 bind to Teads via their TBD to drive expression of target genes. To test whether this interaction is important for their function, we evaluated the *yap1^bns22* zebrafish allele, which encodes an in-frame deletion in the TBD of Yap1 (p.(Pro42_Glu60delinsLeu)) (*Figure 2—figure supplement 1*). Specifically, this deletion spans across the S54 residue that binds to Teads (*Miesfeld et al., 2015*; *Zhao et al., 2008*). In the *yap1^bns22/bns22*;*wwtr1^-/-* embryos, we again observed apoptosis in the epidermis (*Figure 2—figure supplement 1*). These observations suggest that nuclear-localized Yap1 in EBCs promote survival by driving the expression of downstream target genes with Teads.

## Yap1 and Wwtr1 are dispensable for the proliferation of developing epidermal cells

Using adult mice, a previous study found fewer proliferating cells in the epidermis of *yap1;wwtr1* double mutants (*Elbediwy et al., 2016*). Thus, we performed a proliferation assay with phospho-histone H3 (pH3) staining in the 3–5 ss embryos. We chose this developmental stage to avoid coinciding with the cell death phenotype in mutants. Overall, the number of pH3 foci is not noticeably different between WT and mutant embryos (*Figure 3*). Restricting our analysis to the epidermis on the lateral yolk, the number of pH3-positive cells is not significantly different between the two genotypes (*Figure 3B*). Thus, Yap1 and Wwtr1 are not required for epidermal cell proliferation during development.

## Apoptosis in mutant EBCs occurs through the extrinsic pathway

We next address whether the apoptosis phenotype in mutant EBCs is initiated by the intrinsic or extrinsic pathway. Supporting the TUNEL results, we find in mutant epidermis, cleaved Caspase-3 co-stained with P63 and in apoptotic fragments (*Figure 4A and A′*). To test the extrinsic pathway, we evaluated the activity of Caspase-8 using a live Caspase-8 probe, FITC-IETD-FMK (Materials and methods). At the 18–20 ss, FITC foci are present in mutant, but not WT, epidermal cells (*Figure 4B and B′*), indicating that Caspase-8 is active in mutants. This observation indicates that apoptosis in *yap1;wwtr1* double mutant epidermal cells is initiated by the extrinsic pathway.

## γH2AX is recruited to the nuclei of mutant epidermal cells

Previous studies have found that the epidermis clears DNA-damaged cells through the extrinsic apoptotic pathway (*Bang et al., 2003*; *Hill et al., 1999*; *Takahashi et al., 2001*), prompting us to investigate our mutant epidermal cells for DNA damage. Indeed, interrogating mutant and WT embryos by Western blot shows elevated levels of γH2AX, a DNA damage marker, in mutants (*Figure 4—figure supplement 1A*). Closer inspection of these embryos by confocal microscopy reveals a pan-nuclear pattern of γH2AX in mutant EBCs but not in WT ones (*Figure 4C and C′*). γH2AX was also observed in apoptotic fragments. Therefore, DNA damage is the putative stimulus of the extrinsic apoptotic pathway in mutant EBCs.

## Mutant basal cell size growth is limited

What are the cellular events leading to apoptosis in mutant embryos? We utilized live imaging of the developing epidermis in order to visualize the cell behavior. Embryos at the one-cell stage were injected with *H2A-mCherry* and *lyn-EGFP* mRNA to mark the nuclei and membrane, respectively. At the tailbud stage, we mounted the embryos laterally on the yolk to record the epidermal cells. In sibling and *yap1;wwtr1* double mutants, epidermal cell behavior is not readily distinguishable between them throughout the live-imaging experiment, aside from apoptosis in mutant embryos beginning from the 3 hr mark (*Video 1*).

With the membrane marker, we measured basal cell size (basal cell apical area) in sibling and mutant embryos. Whereas basal cells in sibling embryos grew steadily throughout the live-imaging experiment, mutant ones stopped growing 120 min post-tailbud (*Figure 5*). At the end of this experiment, basal cells in sibling embryos were on average a third larger than mutant ones (747±136 µm² vs.

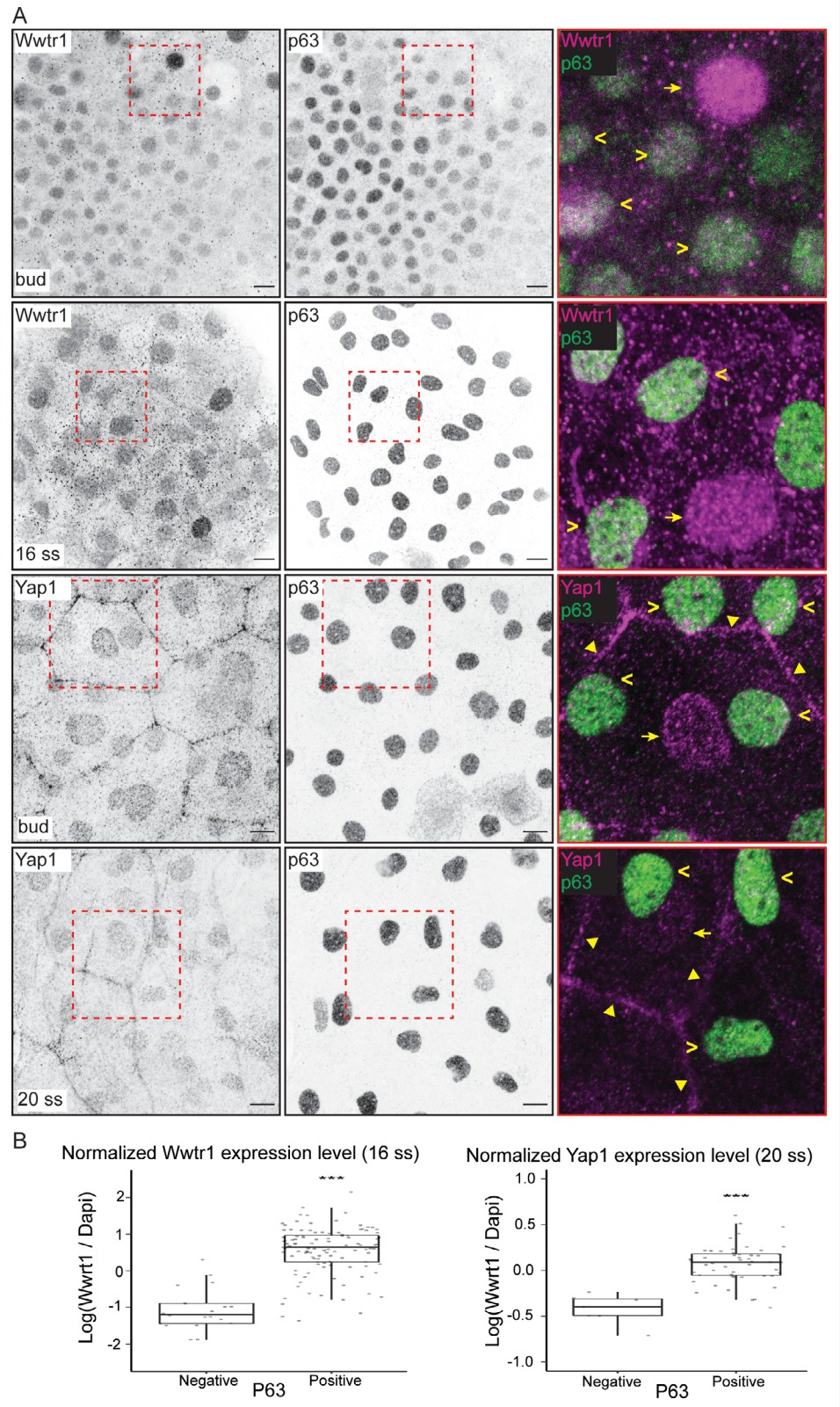

**Figure 2.** Yap1 and Wwtr1 are localized in epidermal basal cells (EBCs) and EVL cells. (**A**) Maximum intensity projections of the epidermis on the lateral yolk stained with P63, and Wwtr1 or Yap1 antibodies at indicated developmental stages. Insets, demarcated in red, show overlay of P63 and Yap1/Wwtr1. Arrowheads – P63-positive basal cells; arrows – peridermal cells; triangles – peridermal cell junctions. Scale bars, 10 µm. (**B**) Boxplots

*Figure 2 continued on next page*

*Figure 2 continued*

of normalized intensities of Yap1 and Wwtr1 in the nucleus of P63-positive and P63-negative cells in the basal epidermis. t-Tests were carried out to compare these intensities between the two groups. ***p<0.001.

The online version of this article includes the following source data and figure supplement(s) for figure 2:

**Figure supplement 1.** Yap1 and Wwtr1 are localized to the nuclei of epidermal basal cells (EBCs), and Yap1's Tead-binding domain is essential in EBC survival.

**Figure supplement 1—source data 1.** Normalized Yap1 and Wwtr1 intensity measurements for *Figure 2—figure supplement 1A*.

---

551±103 μm$^2$ [mean ±s.d.]; p<0.01). Curiously, apoptosis – DNA condensation and membrane blebbing (*Video 2*) – of mutant basal cells follows after stagnation of their growth (*Figure 5*). We investigated other cell shape parameters, but we did not observe any clear differences between mutant and sibling embryo basal cells. Taken together, basal cell apical area size is inhibited in *yap1;wwtr1* double mutant embryos.

## Investigation of γH2AX by cell stretching and substrate stiffness

Our results show that DNA damage is an upstream trigger of extrinsic apoptosis in mutant EBCs, and that apoptosis occurs after inhibition of cell size. Thus, we hypothesized that factors underlying cell size play a role in DNA damage. Exploring this hypothesis in the early embryo is very challenging due to its accessibility for modulating forces. We tested this hypothesis with two experiments: (1) in vivo stretching of head epidermal cells by ventricular inflation and (2) culturing of a human keratinocyte cell line, HaCaT, on increasing stiffness of collagen-coated hydrogels. Though not directly related to EBCs, these experiments allow us to test the link between cell size and DNA damage, and the potential role of the Hippo pathway in regulating this process.

We stretched epidermal cells in vivo by inflating the zebrafish brain ventricle with mineral oil (*Figure 6A*), based on a previously characterized approach (*Lewis et al., 2020*) As a positive control, we exposed zebrafish embryos to UV for 10 min followed by recovery of 1 hr before fixation. In the UV-treated embryos, we found pan-nuclear staining of γH2AX in the periderm and basal epidermis (*Figure 6—figure supplement 1A*), similar to the staining pattern in mutant EBCs (*Figure 4C*). However, we did not observe γH2AX in the nuclei of epidermal cells after ventricular inflation (*Figure 6A'*). These observations suggest that stretching of epidermal cells alone does not induce the recruitment of γH2AX.

Next, to test our hypothesis in defined mechanical environments, we evaluated DNA damage in HaCaT cells cultured on increasing substrate stiffness – 0.5, 12, 25 kPa hydrogels and glass. We used these stiffnesses as they cover a broad range of physiologically relevant conditions, from very

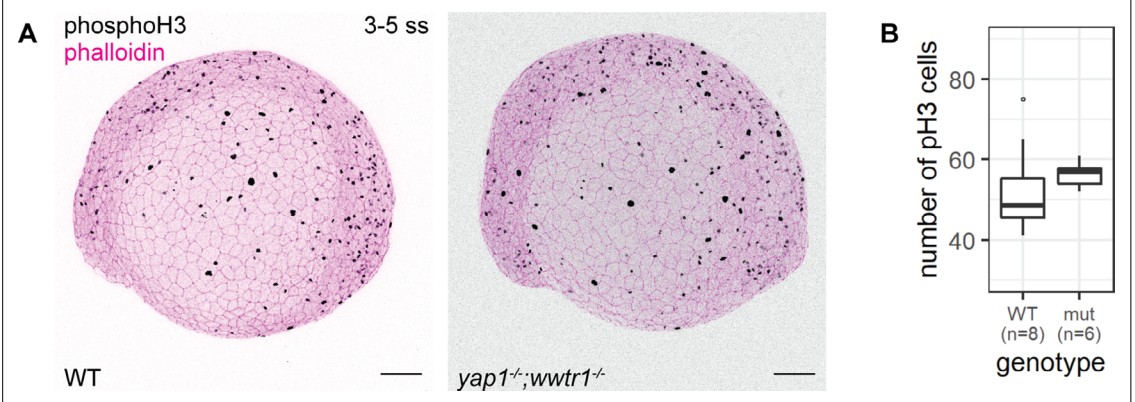

**Figure 3.** Concurrent loss of *yap1* and *wwtr1* did not impair epidermal cell proliferation. (**A**) Maximum intensity projections of 3–5 ss WT and mutant embryos stained with a proliferation marker, phospho-histone H3 (phosphoH3; pH3), and phalloidin. Scale bars, 100 μm. (**B**) Boxplot of the number of pH3 cells on the lateral side of the yolk of WT and mutant zebrafish embryos. mut – *yap1;wwtr1* double mutants.

The online version of this article includes the following source data for figure 3:

**Source data 1.** Phospho-histone H3 (pH3) count data for *Figure 3B*.

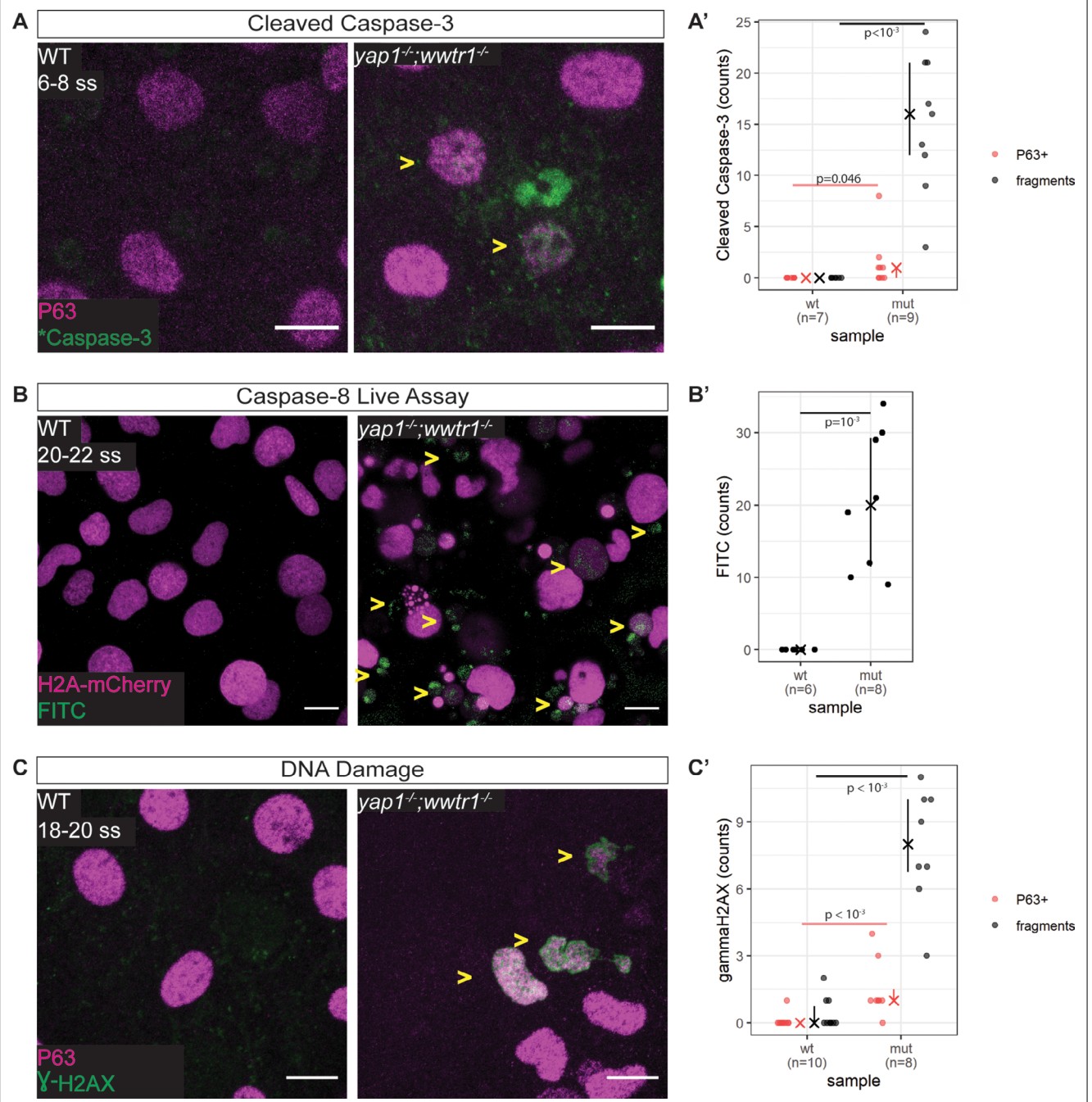

**Figure 4.** Zebrafish *yap1⁻/⁻;wwtr1⁻/⁻* epidermal cells exhibit DNA damage and extrinsic apoptotic cell death. (**A**) Maximum intensity projections of 6–8 ss WT and mutant epidermis on the lateral yolk stained with cleaved Caspase-3 (*Caspase-3) and P63. Some *Caspase-3-positive cells are also P63-positive (yellow arrowheads). Scale bars, 10 μm. (**A'**) Number of epidermal basal cells (EBCs) (P63+) and apoptotic fragments (fragments) expressing cleaved Caspase-3 in WT and mutant embryo epidermis. 'X' represents the median, while the whiskers projecting from it represent the interquartile range. (**B**) Maximum intensity projections of the epidermis on the lateral yolk of 20–22 ss WT and mutant embryos expressing H2A-mCherry. Embryos were incubated in a Caspase-8 chemical probe, FITC-IETD-FMK, prior to imaging. FITC signal indicates Caspase-8 activity in cells (yellow arrowheads). Scale bars, 10 μm. (**B'**) Number of FITC foci in WT and mutant embryo epidermis. 'X' represents the median, while the whiskers projecting from it represent the interquartile range. (**C**) Maximum intensity projections of 18–22 ss WT and mutant epidermis on the lateral yolk stained with γH2AX and P63. Some nuclei are positive for both markers (yellow arrowheads). Scale bars, 10 μm. (**C'**) Number of EBCs (P63+) and apoptotic fragments (fragments) expressing γH2AX in WT and mutant embryo epidermis. 'X' represents the median, while the whiskers projecting from it represent the interquartile range.

The online version of this article includes the following source data and figure supplement(s) for figure 4:

**Source data 1.** Casp3, Casp8, and γH2AX count data for *Figure 4A', B' and C'*, respectively.

*Figure 4 continued on next page*

*Figure 4 continued*

**Figure supplement 1.** Increased γH2AX levels in mutants.

**Figure supplement 1—source data 1.** Raw image files of Western blot assay for *Figure 4—figure supplement 1B*.

soft (0.5 kPa) to very stiff. One set of cells was exposed to UV as a positive control. Interestingly, the number of γH2AX foci detected in HaCaT cell nuclei is higher in cells cultured on stiffer substrates, but this correlation is not linear, with the maximum number of foci occurring in HaCaT cells cultured on 12 kPa hydrogel (*Figure 6B*). Moreover, we observed a similar trend among cells exposed to UV (*Figure 6—figure supplement 1B*). Nevertheless, in cells cultured on a given substrate stiffness, we detected more γH2AX foci in the UV group than the unexposed group (*Figure 6—figure supplement 1B*). Thus, keratinocytes cultured on stiff substrates appear to have greater genomic stress than cells on soft substrates.

As YAP1 and WWTR1 are shuttled into the nucleus when cells are cultured on stiffer substrates, we hypothesized that their presence in the nucleus protects HaCaT cells cultured on 25 kPa and glass from genomic stress resulting in fewer γH2AX foci. We performed knockdowns of *YAP1*, *WWTR1*, or both in HaCaT cells cultured on glass. We found that double knockdown of *YAP1* and *WWTR1* did not deplete their protein expression, unlike the single *YAP1* or *WWTR1* knockdowns (*Figure 7A*). Accordingly, the depletion of YAP1 or WWTR1 resulted in greater expression of γH2AX by Western blot (*Figure 7A*), and more γH2AX foci (*Figure 7B*). However, we could not test the redundancy of YAP1 and WWTR1 as the double knockdown was incomplete, which may explain the unchanged γH2AX levels in the double knockdown compared to control. This observation suggests a role for YAP1/WWTR1 in protecting the genome from DNA damage.

## Discussion

Our results presented here are consistent with Yap1 and Wwtr1 playing important roles in epidermal cell size control, genomic stability, and survival during epidermal development. In contrast to adult murine epidermis (*Elbediwy et al., 2016*), the developing zebrafish epidermis does not require Yap1 and Wwtr1 for cell proliferation. Instead, we found mutant epidermal cells to be smaller, exhibit DNA damage, and undergo apoptosis through the extrinsic pathway. These observations led us to investigate the relationship between the mechanical environment of epidermal cells and genomic stress.

Mechanical perturbations of cells were shown to recruit DNA damage factors including γH2AX and ATR (*Cognart et al., 2020*; *Kumar et al., 2014*). Our present results show that both baseline and induced DNA damage can be intensified by culturing cells on stiffer substrates. Specifically, we found more γH2AX foci in human keratinocyte cells seeded on 12 kPa, 25 kPa hydrogels, and glass compared to the ones seeded on 0.5 kPa hydrogels. At the same time, YAP1/WWTR1 are known to shuttle into the nucleus of cells cultured in stiffer substrates. This phenomenon may explain why the number of γH2AX is fewer in cells cultured on 25 kPa hydrogel and glass than cells cultured on 12 kPa hydrogel, if YAP1 and WWTR1 protect the genome from DNA damage. Indeed, depleting them in HaCaT cells and zebrafish epidermal cells results in greater expression of γH2AX. We note that these conclusions are based on results from different experimental systems, and further work will be required to confirm these ideas.

Yap1 was shown to supply survival signals in cells under genotoxic stress conditions by driving the expression of *Survivin* (*Guillermin et al., 2021*; *Ma et al., 2016*). However, our Western blot assay did not find downregulation of Survivin in zebrafish *yap1;wwtr1* double mutants (data not shown), presumably as Survivin is also a key player in cellular proliferation (*Nair et al., 2013*), which is not impeded in mutants. Therefore, we propose that the mechanical environment of the epidermis

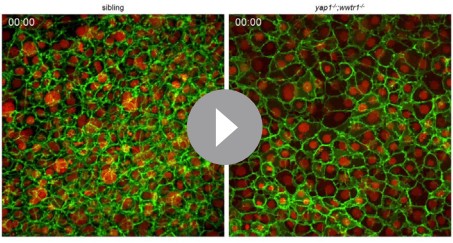

**Video 1.** Time lapse imaging of mutant and sibling epidermis. Maximum intensity projections of the epidermis on the lateral yolk of mutant and sibling embryos. Membranes and nuclei are marked by Lyn-EGFP (green) and H2A-mCherry (red), respectively. Time stamp format, HH:MM; 00:00 is tailbud stage. https://elifesciences.org/articles/72302/figures#video1

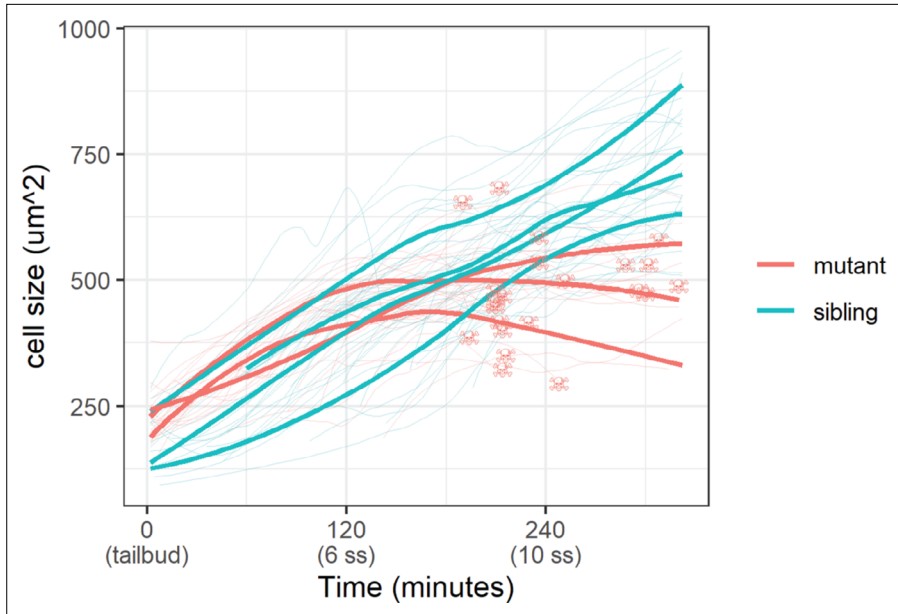

**Figure 5.** Epidermal cell size of sibling and mutant embryos during development. Cell size was measured with a membrane marker during live imaging of mutant and sibling embryos from the tailbud stage. Fine lines are the size of individual cells, while bold lines are the average cell size in a single embryo. Skull symbols mark cell size and time before death (DNA condensation).

The online version of this article includes the following source data for figure 5:

**Source data 1.** Measured cell size from live-imaging experiment for *Figure 5*.

modifies genomic stress in cells, and that Yap1 and Wwtr1 play a role in their survival. However, further work will be required to test this proposal in vivo, by altering the mechanical environment within the embryo.

The mechanical environment of the developing epidermis changes with increasing production of the ECM. Indeed, greater ECM density is known to increase cell size (*Engler et al., 2006*), which could underscore the increasing basal cell size during epidermal development. In the transcriptomic signatures from the previous study (*Kimelman et al., 2017*), genes that are downregulated in *yap1;wwtr1* double mutant tailbuds include *lama5*, *col7a1*, *lamb1a*, *lamb4,* and *lamb2*, which encode key components of the skin basal lamina. This downregulation likely translates to a different ECM environment in mutant embryos and consequently results in the inhibition of basal cell size (*Figure 5*).

Our work reveals that Yap1 and Wwtr1 likely have important functions in epidermal cell survival and growth during development. We also show greater genomic stress in cells grown on more rigid extracellular substrates, which is consistent with the DNA damage-induced apoptosis phenotype in *yap1;wwtr1* double mutant zebrafish. These results open new areas of exploration on the physiological relevance of mechanobiology in development, physiology, and disease, as well as its interplay with Yap1/Wwtr1 and genomic stress in these processes.

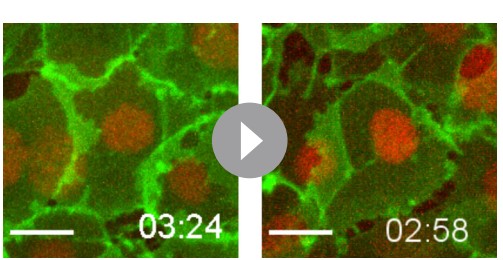

**Video 2.** Apoptosis of mutant basal cells. Time lapse of mutant basal cells undergoing apoptosis as captured by live imaging. Membranes and nuclei are marked by Lyn-EGFP (green) and H2A-mCherry (red), respectively. Time stamp format, HH:MM; 00:00 is tailbud stage. Scale bars, 10 μm.

https://elifesciences.org/articles/72302/figures#video2

## Materials and methods
### Zebrafish lines
Mutant zebrafish lines used in this study are: *yap1[bns19]* (*Astone et al., 2018*; *Kimelman et al., 2017*), *yap1[bns22]* (this study), and *wwtr1[bns35]* (*Lai et al., 2018*).

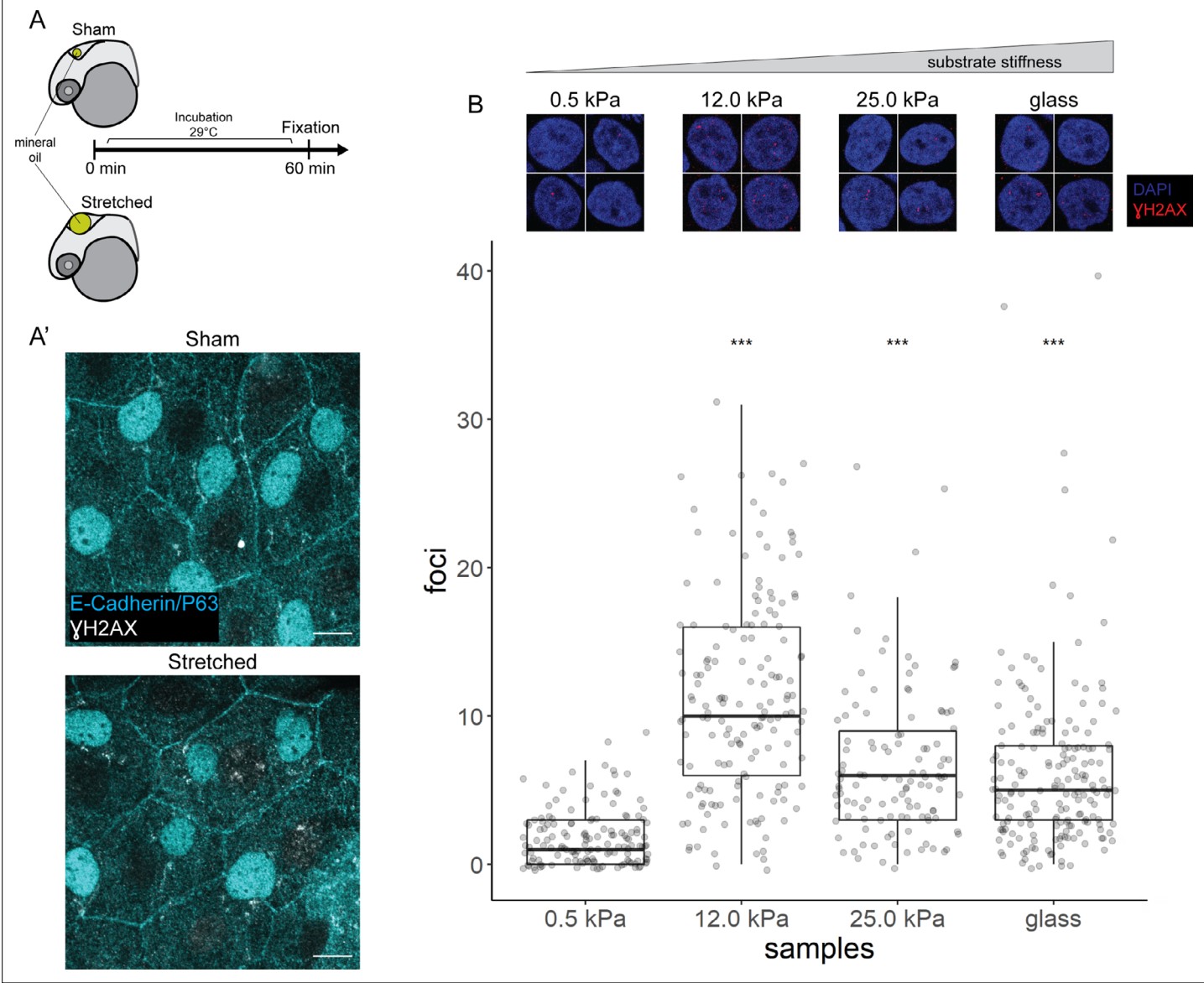

**Figure 6.** γH2AX in stretched epidermal cells and in keratinocytes cultured on different substrate stiffness. (**A**) Schematic of head epidermal cell stretching experiment in zebrafish embryos. (**A'**) Sham (n=6) and stretched (n=6) head epidermal cells stained with γH2AX and epidermal markers (E-cadherin and P63). P63 is a marker for epidermal basal cells (EBCs). No discernible γH2AX signal was detected in both conditions. Scale bars, 10 μm. (**B**) Selected nuclei of HaCaT cells cultured on 0.5, 12.0, 25.0 kPa hydrogels and glass, as well as boxplot of the number of γH2AX foci in these nuclei. Number of γH2AX foci in HaCaT cell nuclei were contrasted against HaCaT cells cultured on 0.5 kPa hydrogel using t-tests. Data were collected from three independent experiments. ***p<0.001 adjusted for multiple testing (Tukey method).

The online version of this article includes the following source data and figure supplement(s) for figure 6:

**Source data 1.** γH2AX foci count data for *Figure 6B*.

**Figure supplement 1.** γH2AX in zebrafish head epidermis and HaCaT cells exposed to UV.

**Figure supplement 1—source data 1.** γH2AX foci count data for *Figure 6—figure supplement 1B*.

The *yap1<sup>bns22</sup>* allele was generated from a CRISPR/CAS9-mediated mutagenesis. Two single-guide RNA (sgRNA) sequences, 5'-ACCTCATCGGCACGGAAGGG-3' and 5'-CTGGAGTGGGACTTTG GCTC-3', were designed by an sgRNA design tool by the Zhang Lab (crispr.mit.edu; accessed in 2014). These sequences were cloned into the pT7-gRNA vector (Addgene #46759). Following vector linearization with BsmBI enzyme, sgRNA were obtained by in vitro transcription with MEGAshortscript T7 kit (Ambion). *CAS9* mRNA was obtained by in vitro transcription of linearized pT3TS-nCas9n vector

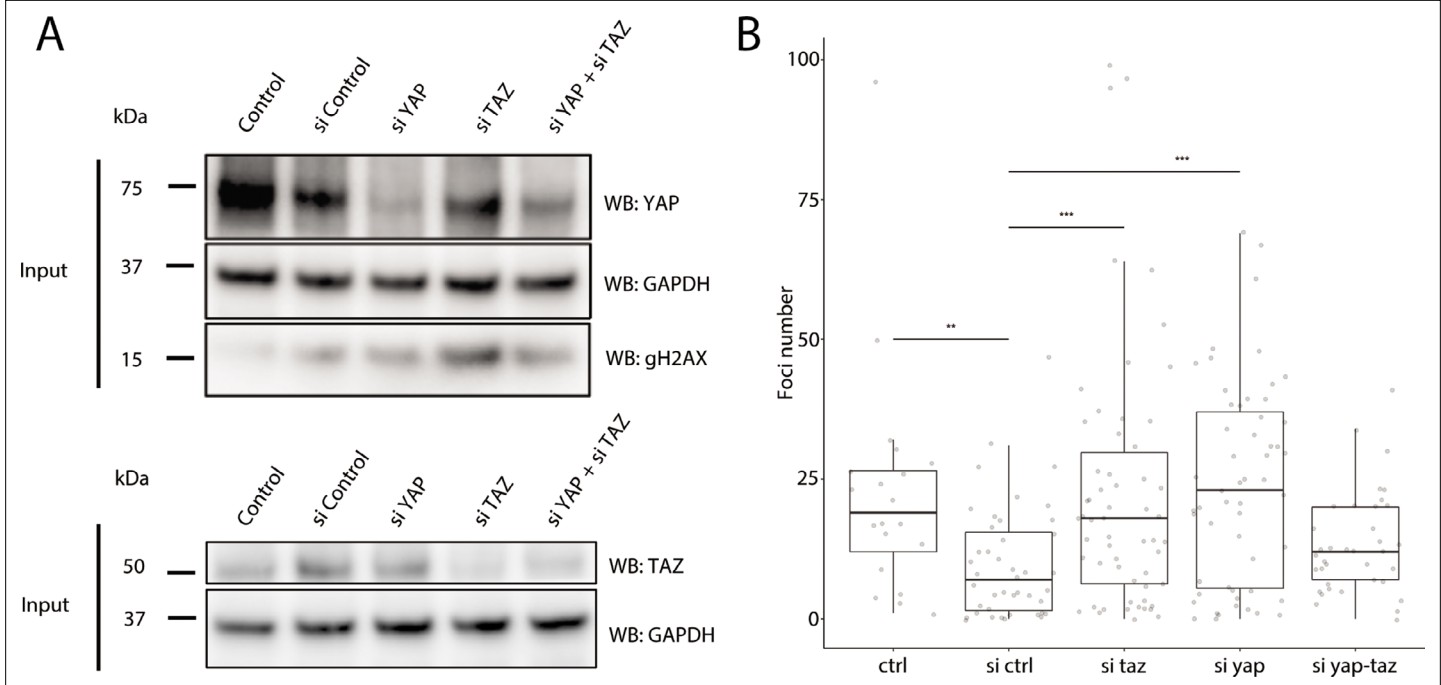

**Figure 7.** Recruitment of γH2AX in *YAP1/WWTR1* knockdown HaCaT cells. (**A**) Western blots of control and *YAP1/WWTR1* knockdown HaCaT cells. (**B**) Boxplot of the number of γH2AX foci in control and *YAP1/WWTR1* knockdown HaCaT cell nuclei. Number of γH2AX were compared to control knockdowns (sictrl) using t-tests.

(Addgene #46757) with MEGAscript T3 kit (Ambion). sgRNA (50 pg each) and 150 pg of *CAS9* mRNA were co-injected into one-cell stage AB embryos. The *yap1^{bns22}* allele contains a 54 bp deletion in the TBD [p.(Pro42_Glu60delinsLeu)]. This allele can be genotyped by using the following PCR primer pair: 5'-CTGTTTGTGGTTTCTGAGGGG-3' and 5'-CGCTGTGATGAACCCGAAAA-3'. Mutant and WT products can be resolved and distinguished by gel electrophoresis with a 2% gel. Genotyping of *yap1^{bns19}* and *wwtr1^{bns35}* alleles have been described previously (*Astone et al., 2018*; *Lai et al., 2018*).

Mutant embryos in this study were obtained from an incross of double heterozygous animals (i.e. *yap1^{bns19/+}*;*wwtr1^{bns35/+}* incross or *yap1^{bns22/+}*;*wwtr1^{bns35/+}* incross).

## Incubation and visual identification of mutant embryos

Experiments involving 16 ss and older embryos were incubated at 25°C throughout, otherwise younger embryos were incubated at 28.5°C. *yap1;wwtr1* double mutant embryos older than 16 ss can be visually identified for experiments. These mutant embryos were compared to stage-matched embryos from WT crosses.

For experiments involving <16 ss embryos, embryos from heterozygous incross were fixed and assayed for cell death with TUNEL. Using a wide-field stereomicroscope, embryos with excess TUNEL staining in the epidermis on the lateral yolk were selected and mounted for confocal imaging. Under the confocal microscope, embryos with apoptotic bodies (TUNEL foci and fragmented DNA stained by DAPI) were further investigated and genotyped for downstream analyses. These embryos were compared to stage-matched WT embryos.

## Whole-mount immunofluorescence assay

Zebrafish samples were fixed in 4% PFA in 1× PBS, followed by dechorionation. Briefly, samples were blocked for 1 hr in 2 mg/mL BSA and 1% goat serum (Thermo Fisher) in 1× PBSTXD (1× PBS, 0.1% Tween-20, 0.1% Triton X-100%, and 1% DMSO). Then, samples were incubated in primary antibody (see below) overnight at 4°C. Samples were washed in 1× PBSTXD every 15 min for 2 hr and then incubated in secondary antibody in room temperature for at least 4 hr, followed by washing with 1× PBSTXD. Counter stains for nuclei was DAPI, and for actin was phalloidin (Alexa 488 conjugated).

When using anti-γH2AX as primary antibody, all PBS buffers were substituted with TBS buffers, and blocking buffer consisted of 3–5% BSA only.

TUNEL assays were performed between the blocking step and primary antibody incubation step. Samples were washed in 1× PBST (1× PBS, 0.1% Tween-20) three times, 5 min each. After aspiration of the buffer, samples were incubated in the Fluorescein TUNEL kit (Roche) for 1 hr at 37°C with gentle shaking. 1× PBST washes were performed to stop the reaction before proceeding with the remaining step of the immunofluorescence protocol.

Primary antibodies used in this study were: anti-Yap1 (CS4912, Cell Signaling), 1:250; anti-Wwtr1 (D24E4, Cell Signaling), 1:250; anti-P63 (ab735, Abcam), 1:500; anti-ZO1 (1A12, Thermo Fisher), 1:250; anti-phosphoH3 (9701, Cell Signaling), 1:500; anti-cleaved-Caspase-3 (ab13847, Abcam), 1:250; and anti-γH2AX (GTX127342, Genetex), 1:500, anti-E-cadherin (610182, BD Transduction Labs), 1:100.

Secondary antibodies used in this study were: Alexa fluorophore-conjugated anti-mouse and anti-rabbit by Thermo Fisher used at 1:500 dilution.

## Caspase-8 assay

Embryos were injected with 10–15 pg *H2A-mCherry* mRNA at the one-cell stage. At 16 ss, mutant and WT embryos were identified and dechorionated, then incubated in FITC-IETD-FMK (ab65614, Abcam) for 30 min at room temperature. Embryos were washed with egg water three times, 5 min each, and then mounted for confocal imaging.

## Proliferation assay

Embryos were imaged from the lateral side. After imaging, embryos were genotyped and only *yap1;wwtr1* double mutants were compared with stage-matched WT embryos. A maximum intensity projection was generated for each embryo. The yolk was manually demarcated as a 'region of interest', followed by manual identification and counting of pH3 foci. Poisson regression for count data was used to test whether the number of pH3 foci differs between WT and mutants.

## Yap1/Wwtr1 immunostaining quantification

Nuclei were segmented with DAPI in order to get total intensity of Yap1, Wwtr1, and P63. Peridermal nuclei were excluded from the analysis. The intensity of interests are normalized with DAPI intensity. Nuclei were grouped as P63-positive and P63-negative. t-Tests were performed between the P63-positive and P63-negative groups for each normalized Yap1 and Wwtr1 intensity.

## Confocal imaging

Live and fixed embryos were embedded in 1% LMA (Sigma) on glass-bottomed dish (inverted microscopes) or Petri dish (upright microscopes). For low-magnification images of the entire embryo, a 10× objective lens was used. Otherwise, high-magnification images were taken with 40× objective (water dipping) or 63× objective (water immersion) lens with at least 1 NA was used. For the ventricular inflation experiment, the stretched region of the head epidermis was flattened and mounted using a coverslip and imaged on the Zeiss LSM 880 confocal microscope with a Plan-APOCHROMAT 63×/1.4 oil objective at a 1.5× optical zoom.

## Live imaging

Embryos from incrosses of *yap1;wwtr1* double heterozygous animals were injected with 10 pg of *H2A-mCherry* and *lyn-EGFP* mRNA each at the one-cell stage. At 90% epiboly, 6–10 embryos were dechorionated and embedded laterally in 0.5% LMA on a glass-bottomed dish to be imaged under the spinning disk system (Yokogawa CSU-W1). A z-stack of 41 slices, 0.5 μm thick, were acquired per embryo every 2 min overnight in an environmental chamber set at 28.5°C. At most eight embryos were imaged in a single sitting. Embryos were genotyped the next day. All mutants are $yap1^{-/-};wwtr1^{-/-}$, while siblings are not $yap1^{-/-}$ or $wwtr1^{-/-}$.

## Cell size quantification

Maximum intensity projections of each embryo were generated for each time point. Basal cells were distinguished from EVL cells manually. The lyn-EGFP marks the cell membrane which was manually

demarcated to measure cell size every five frames. When a cell divides, it marks the start or end of the tracking. For dying cells in mutants, the tracking stops one frame before the DNA condenses.

## Western Blot

16–18 ss mutant embryos (visually identified) and stage-matched WT embryos were dechorionated and collected in separate Eppendorf tubes. A total of 20 embryos from each group were pooled into one tube to constitute a biological replicate. In 1× PBS, embryos were mechanically de-yolked with a pipette tip. Samples were washed in cold 1× PBS thrice, and then 4× Laemmli sample buffer (Bio-Rad) with B-mercapthoethanol (Sigma) was added. Samples were vortexed and boiled at 95°C for 5 min three times. Samples were stored in –20°C until ready for use.

Proteins from each sample were resolved using a precast 8–16% gradient gel (Bio-Rad), and transferred to a PVDF membrane following the manufacturer's protocol. Membranes were blocked with 3–5% BSA in 1× TBST (1× TBS, 0.5% Tween-20) for an hour, followed by incubation with primary antibody overnight at 4°C. Membranes were washed with 1× TBST before incubation with secondary antibody for at least 1 hr. Membranes were washed again with 1× TBST. ECL substrate (Bio-Rad) was added to the membrane and imaged with the ChemiDoc imaging system (Bio-Rad).

The primary antibodies used in this assay are: anti-B-actin (A5441, Sigma), 1:1000; anti-γH2AX (GTX127342, Genetex), 1:1000.

The secondary antibodies used in this assay are: HRP-conjugated anti-mouse (31430, Thermo Fisher), 1:10,000; HRP-conjugated anti-rabbit (31460, Thermo Fisher), 1:10,000.

## Ventricular oil injections

The head epidermis was stretched by injecting hydrophobic mineral oil (Bio-Rad; 1632129) into the hindbrain ventricle of 24 hpf zebrafish, as previously described (*Lewis et al., 2020*). In brief, at 24 hpf the dechorionated embryos were anesthetized in 0.02–0.024% (w/v in E3) tricaine, and vertically mounted in holes punched in a 3% (w/v) agarose plate. This was followed by either stretching the epidermis by injecting 10–12 nL mineral oil in the hindbrain ventricle (stretched) or injecting only 0.1–0.2 nL oil to act as an injury control (sham). All embryos used in a set were obtained from the same clutch and were injected using the same capillary needle, to reduce variation. The test and sham embryos were grown in E3 until fixation an hour later.

## HaCaT cell culture

All cell culture reagents were purchased from Gibco (Thermo Fisher Scientific, Waltham, MA). The human immortalized keratinocyte cell line, HaCaT, was a gift from CT Lim Lab (MBI, Singapore). The core facility at the Mechanobiology Institute, National University of Singapore regularly performs mycoplasma contamination checks on these cells. All experiments on these cells were performed within 1 year of receiving them. HaCaT keratinocytes were cultured in complete Dulbecco's modified Eagle's medium that was supplemented with 100 IU/mL aqueous penicillin, 100 µg/mL streptomycin and 10% heat-inactivated fetal bovine serum. Cells were maintained at 37°C in a humidified atmosphere containing 5% $CO_2$ and harvested with TrypLE (1×). TrypLE was deactivated with complete media. Cells were pelleted by centrifugation at 180× *g* for 5 min and then re-suspended in complete media. The cell suspensions were used only when their viability, as assessed by trypan blue exclusion, exceeded 95%. Briefly, 0.1 mL of 0.4% of trypan blue solution was added to 0.1 mL of cell suspension. Cell viability and density were calculated with a hemocytometer.

HaCaT cells were then seeded on 0.5, 12, and 25 kPa collagen-coated Softslips (Matrigen, Brea, CA), as well as glass coverslips in individual wells of a 24-well plate. 50,000 cells were seeded on each substrate. Seeded cells were incubated for 10 hr or overnight for cells to completely attach to the substrates.

## HaCaT cell immunofluorescence assay

HaCaT cells were fixed with 4% PFA in 1× PBS for 15 min. After washing off residual PFA with 1× TBSX (1× TBS supplemented with 0.1% Triton X-100), samples were incubated with blocking solution (4% BSA in 1× TBSX) for 1 hr. Samples were then incubated in primary antibody diluted in blocking solution overnight at 4°C. After washing off the primary antibody solution with 1× TBSX, samples were incubated in secondary antibodies and counterstains diluted in blocking solution for 2 hr, followed

by washes with 1× TBSX. Hydrogel-coated coverslips (Softslips) were inverted unto a coverslip for imaging according to the manufacturer's instructions.

## γH2AX foci quantification in HaCaT cells

Nuclei of HaCaT cells were marked as regions of interest for counting the number of γH2AX foci. Background signal was subtracted using a rolling radius of 5 pixels. A focus is at least 5 pixels in area size, and the number of foci was counted for each nucleus. To test the increase of γH2AX foci in the UV-treated group, one-sided t-tests were performed between UV and control groups for each substrate stiffness. To test the change of the number of γH2AX foci by different substrate stiffness, control HaCaT cells cultured on 0.5 kPa hydrogels were used as the reference in two-sided t-tests. p-Values were adjusted for multiple hypothesis testing with the Tukey method.

## UV exposure

Twenty-four hpf zebrafish embryos or HaCaT cells were exposed to UV in the Biosafety Cabinet. UV dose for zebrafish experiment was 63.2 mJ/cm$^2$, while UV dose for HaCaT cells was 30.6 mJ/cm$^2$. After UV exposure, samples were allowed to recover for an hour (zebrafish in a 29.0°C incubator; HaCaT cells in 37.0°C humidified incubator).

## Statistical analysis

Paired t-test analysis was performed in *Figure 4* using estimationstats.com (*Ho et al., 2019*).

## Acknowledgements

The *yap1*[bns22] zebrafish allele was generated by JKHL in Didier Stainier's lab (MPI, Bad Nauheim), and we thank him for his support. We also thank Boon Chuan Low (MBI, Singapore), Mahendra Sonawane (TIFR, India), and Tom Carney (NTU, Singapore) for insightful discussions and suggestions. This work was funded by the Singapore Ministry of Education Tier 3 grant MOE2016-T3-1-002 and Mechanobiology Institute core funding.

## Additional information

### Funding

| Funder | Grant reference number | Author |
| --- | --- | --- |
| Ministry of Education - Singapore | MOE2016-T3-1-002 | Jason KH Lai<br>Pearlyn JY Toh<br>Hamizah A Cognart<br>Timothy E Saunders |

The funders had no role in study design, data collection and interpretation, or the decision to submit the work for publication.

### Author contributions

Jason KH Lai, Conceptualization, Data curation, Formal analysis, Investigation, Methodology, Validation, Visualization, Writing – original draft, Writing – review and editing; Pearlyn JY Toh, Data curation, Investigation, Writing – review and editing; Hamizah A Cognart, Geetika Chouhan, Data curation, Investigation; Timothy E Saunders, Conceptualization, Formal analysis, Funding acquisition, Methodology, Resources, Supervision, Writing – review and editing

### Author ORCIDs

Jason KH Lai http://orcid.org/0000-0002-3476-4733
Pearlyn JY Toh http://orcid.org/0000-0003-0907-7947
Hamizah A Cognart http://orcid.org/0000-0002-3090-1526
Timothy E Saunders http://orcid.org/0000-0001-5755-0060

### Ethics

Ethics statementAll zebrafish husbandry was performed under standard conditions in accordance with institutional (Biological Resource Center, A*Star, Singapore, and Tata Institute of Fundamental

Research, India) and national ethical and animal welfare guidelines (Singapore IACUC: 181,323 and GMAC: Res-21–034). All users were trained in ethical handling of zebrafish.

### Decision letter and Author response
Decision letter https://doi.org/10.7554/eLife.72302.sa1
Author response https://doi.org/10.7554/eLife.72302.sa2

## Additional files

### Supplementary files
• Transparent reporting form

### Data availability
All data generated or analysed during this study are included in the manuscript and supporting file; Source Data files have been provided for Figures 1, 3, 4, 5 and 6.

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
