## [Editor Report]

This manuscript identifies a novel function for YAP/TAZ in epithelial cells. Mechanical stress/strain is shown to drive genomic stress and a DNA damage response in zebrafish skin, and concurrent activation of YAP/TAZ is essential to accommodate the stress/strain by allowing elastic expansion and flattening of cells as well as preventing apoptosis.

---

## [Decision Letter]

**Decision letter after peer review:**

Thank you for submitting your article "DNA-damage induced cell death in *yap1;wwtr1* mutant epidermal basal cells" for consideration by *eLife*. Your article has been reviewed by 3 peer reviewers, and the evaluation has been overseen by a Reviewing Editor and Marianne Bronner as the Senior Editor. The following individual involved in review of your submission has agreed to reveal their identity: Barry Thompson (Reviewer #3).

The reviewers have discussed their reviews with one another, and the Reviewing Editor has drafted this to help you prepare a revised submission. While the reviewers think that manuscript is interesting and has the potential to make an important contribution to the literature, they also pointed out several issues and clarifications that were necessary. The major points are summarized in Essential Revisions below but we have also included the full reviews to help you better understand the issues raised.

Essential revisions:

The most important point seems that there is a lack of direct evidence that the cell size/cell stretching phenotype and the apoptosis phenotype are due to mechanical stress. There needs to be better evidence to support this.

Is it known whether different regions of the epidermis experience different levels of mechanical stress? If so, there could be significant differences in mechanical stress among different regions of the epidermis.

The authors need to consider cell area, which they measure, versus cell volume.

How dependent is periderm growth on proliferation of the EBCs at this stage of zebrafish development? The authors should comment this point.

There was some confusion from the reviewers because you mention in the abstract previous work on the ventral fin while that region was not examined in this study. This is an example of where the broader audience would benefit from some further details/text changes

*Reviewer #1 (Recommendations for the authors):*

Yap1 and Wwtr1 are important, partially redundant proteins involved regulation of organ size and have been extensively studied. Yap1 and Wwtr1 are known to prevent apoptosis. In the adult mouse epidermis, Yap1/Wwtr1 regulate cell proliferation during wound healing. The role of Yap1/Wwtr1 in epidermis development is unknown and the topic of this study.

The manuscript by Lai et al., focuses on the role of Yap1/Wwtr1 in the development of the zebrafish epidermis. The authors present convincing data that epidermal basal cells (EBCs) exhibit reduced cellular growth and increased DNA damage and apoptosis in Yap1/Wwtr1 double mutant zebrafish embryos. The major weakness of the study is the underlying mechanisms of these phenotypes is not clearly established.

The data presented in the first five figures is convincing. Yap1/Wwtr1 are necessary for the increase in cell size of EBCs and prevent apoptosis of these cells in the post-gastrulation zebrafish embryo. The authors hypothesize that increasing mechanical stress on the Yap1/Wwtr1 mutant EBC cells causes the apoptosis.

In figure six, the authors try a nice in vivo experiment in which they increase mechanical stress in the head of the developing zebrafish embryo by injecting a bolus of mineral oil into the brain. However, this manipulation did not result in the expected increase in apoptosis. It is presumed that this experiment was performed in wild-type embryos as the authors do not specify the genotype. It is not clear why the authors did not try this experiment in Yap1/Wwtr1 mutant embryos as they should be more sensitive to this manipulation according to the author's model.

In the latter part of figure 6, the authors plate human keratinocyte cells on substrates of different stiffnesses. Here, they observe an increase in DNA damage on cells plated on substrates with a stiffness {greater than or equal to}12.0 kPa. These keratinocytes are also prone to DNA damage via UV irradiation, but it doesn't appear that plating cells on stiffer substrates make them more sensitive to UV induced DNA damage. The relevance of these keratinocyte experiments to understanding the phenotype of Yap1/Wwtr1 mutants is not entirely clear.

The major weakness of the study is the lack of a mechanistic understanding of the cell growth and apoptosis phenotypes. In the discussion, the authors hypothesize that the defect in cell growth is due to a reduction in the density of the ECM. This is based on the observations that ECM density has been shown to increase cell size in other contexts and the fact that the expression of a number of ECM proteins is reduced in Yap1/Wwtr1 mutants. This hypothesis is easily testable via genetic mosaics as one would expect this phenotype to be cell non-autonomous. Wild-type EBCs in Yap1/Wwtr1 hosts should also exhibit a reduction in cellular growth. Conversely, Yap1/Wwtr1 EBCs in a wild-type host should exhibit restored cell growth.

The authors suggest that the increase in the ECM and the corresponding increase in the mechanical stiffness of the embryonic environment causes Yap1/Wwtr1 mutant EBCs to apoptose. In this interpretation, which seems to contradict the first hypothesis, Yap1/Wwtr1 mutants do not produce enough ECM for EBCs to reach their normal size, but Yap1/Wwtr1 mutants do produce enough ECM-dependent stiffness to induce EBC apoptosis. The suggested genetic mosaics experiments could also provide evidence for the author's explanation of the apoptosis phenotype. Might one expect an even higher rate of apoptosis of Yap1/Wwtr1 mutant EBCs when they are developing in a more mechanically robust wild-type host?

One issue with the text is ambiguity about what the authors mean by "stress". As in physics, this could mean force per unit area. Alternatively, it could refer to a challenging physiological condition as the term is frequently used in biology. Since this study involves mechanobiology, the authors would need to clearly define what they mean by stress. This ambiguity makes it difficult to always discern the meaning of the text, and it appears that the different meanings are conflated in the text.

Relatedly, it is unclear what the authors mean by "genomic stress". Is the genome under greater mechanical stress in Yap1/Wwtr1 mutants? It would also be helpful to state how the authors think increased substrate stiffness increases genomic stress. Is this relationship purely mechanical or is cell signaling and gene expression involved? What is currently known about the link between the mechanical environment and genomic stress?

The study is interesting and rigorously performed. However, the authors need to better substantiate their conclusions to merit publication in *eLife*. Specifically, the authors need to provide a better understanding of the mechanisms underlying the cell growth and apoptosis phenotypes. The beforementioned mineral oil injections into Yap1/Wwtr1 mutants could provide useful data. However, the genetic mosaic experiments could really be clarifying, and these are standard experiments in zebrafish.

*Reviewer #2 (Recommendations for the authors):*

Lai et al. submit a manuscript investigating the role for Wwtr1/Taz and Yap1 in morphogenesis of the posterior body and epidermal fin fold. Through Zebrafish double mutant embryos, the authors present data suggesting that Wwtr1 and Yap1 cooperate to protect epidermal basal cells from DNA damage during morphogenesis, confirming previously published studies. Not only are both Wwrt1 and Yap1 localised to EBC nuclei, but these cells undergo Tp53 independent apoptosis driven by DNA damage. Recruitment of DNA damage marker γH2AX in mutant embryos is also associated with reduced cell size. Using an in vivo ventricular inflation assay and in vitro cell stretching, the authors suggest that increased tissue stiffness may cause DNA damage that is protected against by Wwrt1 and Yap1.

The submitted manuscript aims to investigate Wwrt1 and Yap1 in epidermal morphogenesis and presents data suggesting a surprisingly strong need for both proteins to protect Epidermal Basal Cell DNA during somitogenesis. While the authors confirm no proliferation defect in mutant embryos, proliferation is assayed early in the mutant pathology and may not represent the cell dynamics present in later stages where the authors also find EBCs to be smaller. A conflict arises between these data as increased cell death and no significant increase in proliferation might lead to epithelial rupture. Why mutant EBC are smaller is not directly addressed but the authors suggest that embryonic stiffening might underlie epidermal susceptibility to DNA damage, however, there are insufficient control experiments and discussion to make this conclusion.

Further experiments and discussion are required to make some of the claims presented here and to improve impact of the work within this hotly studied pathway.

1. Results section is not very well motivated. For example, why 16-18 ss? Not entirely clear whether this stage was analysed by previous publication, why exactly that is so important for the phenotype – why that might be crucial in epidermal development as opposed to other stages.

2. The text states no apoptosis was seen in the tail as before, however, the image shown in Figure 1 does appear to have TUNNEL staining in the posterior embryo albeit much less than the rest of the embryo. Is there any way to quantify this phenotype and represent the average phenotypes for the brood?

3. No justification for studying 6 ss embryos for detailed looked at EBCs. Is there a constant level of apoptosis across these stages of development or does it change over time?

4. Figure 1: Single colour panels would be required to show nuclear TUNNEL staining.

5. Figure 2: Throughout text and legends figure 2 is said to show Wwrt1 and Yap1 expression but only immunofluorescence is shown. Quantification of immunofluorescence should be added to the main figure to bolster the imaging data.

6. Figure 2 S1: The pattern of apoptosis seen in the TEAD binding mutant is different from that seen in the complete null. Could these differences indicate a spatial difference in Wwrt1/Yap1 function? This is not addressed and could be improved by quantification/comparison with other mutants.

7. Why proliferation was assayed in mutants at 3-5 ss is not well justified. Ideally 6 ss and 18 ss would be included to confirm no change in proliferation. This is especially important as not only do mutant cells become smaller, then die more often. How enough cells would be made to compensate for cell loss without rupture of the epithelium leaves these data looking conspicuous and should also be addressed in the text as well.

8. Live Capase-8 probe is not cited in results text.

9. The role for Tp53 in the embryo/cell or why it is used to distinguish the extrinsic apoptotic pathway is not explained.

10. Figure 4: No statistical tests on image quantifications.

11. Cell size was calculated as the 2D area of EBCs, however this is totally cell size as these cells could take on another shape, becoming more columnar. In this case cell size could be the same but tissue geometry would be different and put different force upon the nucleus.

12. Further to comment 7. How can mutant EBCs be smaller and there be no rupturing of the epithelium? Are other parts of the epithelium proliferating more/less to allow for this constraint?

13. It is not clear which EBCs were assayed in ventricular injection experiments. The paper is focussed on the posterior embryo, yet assays whether the head epithelium is susceptible to stretch induced DNA damage. While a potential proxy the strengths and weaknesses of this experiment is not provided – which include the very short time scale of stretching. Further, no data are presented that demonstrate the efficacy of this experiment here.

14. Why gels of these stiffnesses were used has not been justified, nor has the susceptibility of these nuclei to damage at 12Kpa increased compared to greater stiffness been well discussed. How do the collagens within the gels compare to the ECM that EBCs are in contact with? How does the stiffness of the posterior embryo compare to these gels? etc.

15. The claims of the paper would need to be toned down throughout the manuscript without further controls being added.

*Reviewer #3 (Recommendations for the authors):*

This important study identifies a new type of anti-apoptotic function for YAP/TAZ proteins in zebrafish. While YAP/TAZ proteins are known to protect cells from apoptosis in many different animal species, the authors identify a novel and specific requirement for YAP/TAZ in preventing apoptosis caused by mechanical forces that flatten the cell nucleus and lead to DNA damage (an inducer of apoptosis). Double yap1, taz/wwtr1 knockout zebrafish embryos reveal an essential function of YAP/TAZ in maintaining cell survival in the basal layer cells of the epidermis, cells that experience significant mechanical stretch during embryonic morphogenesis. It is proposed that mechanical stress/strain in the epidermis leads to genomic stress, which then requires nuclear translocation of YAP/TAZ to ensure cell survival. In addition, YAP/TAZ are shown to promote elastic expansion and flattening of epidermal cells in response to stretch. The findings are an important addition to previous work in mice, where YAP/TAZ have a mainly pro-proliferative role in basal layer skin epidermal cells during development. Thus, in different contexts, mechanical stress/strain can induce nuclear YAP/TAZ to drive either increased cell proliferation (as in mouse skin), or cell expansion/flattening and survival (as in fish embryonic skin), to govern how the tissue responds to the mechanical stimulus.

The conclusions are clearly supported by the results.

The major strengths are the quality of the zebrafish genetics and phenotypic analysis, as well as the novel insights produced, particularly that the response of zebrafish skin to mechanical stretch is different to that of mouse skin, yet still governed by YAP/TAZ. That mechanical stress leads to genome stress is also a novel finding, and the role of YAP/TAZ in ensuring cell survival in this particular context is also a novel function for these proteins.

These findings will be of major significance to both developmental biology and cancer.

[Editors' note: further revisions were suggested prior to acceptance, as described below.]

Thank you for resubmitting your work entitled "DNA-damage induced cell death in *yap1;wwtr1* mutant epidermal basal cells" for further consideration by *eLife*. Your revised article has been evaluated by Marianne Bronner (Senior Editor) and a Reviewing Editor.

The manuscript has been improved but there are some remaining issues that need to be addressed, as outlined below:

1. The major deficiency of the original submission of this manuscript was a lack of direct evidence that the cell size/cell stretching phenotype and the apoptosis phenotype in yap1/wwtr1 double mutant zebrafish are due to mechanical stress. The authors have addressed this issue not with new data, but by making their conclusions more circumspect. The problem is that the bold conclusions that made the study more broadly interesting are not adequately supported by the data. For example, the impact statement for the manuscript, "Substrate rigidity modulates genomic stress in the epidermal basal cells of the developing zebrafish embryo and Yap1 and Wwtr1 are required for its survival" is not clearly demonstrated by the data in the manuscript. Thus, the study in its current form is of less general interest and may be better suited for a different journal.

2. The author's response is that further experiments are not really feasible. However, this is not entirely true. Genetic mosaic experiments are routine in zebrafish. It is true that transplantation of yap1/wwtr1 double mutant cells into wild-type embryos is challenging because only 1/16 donor embryos will be double mutant, and one doesn't know the genotype of the embryos at the time of transplantation. There are ways to perform the experiment by allowing the donor embryos to develop along with the associated mosaic embryos created by the transplantation. The double mutants could be identified as the phenotype becomes apparent and the associated mosaic embryos could be characterized in detail. Thus, this experiment is difficult, but not impossible.

3. The converse experiment of transplanting wild-type cells into the double mutants is quite doable. Such an experiment could be done to test whether the reduction in ECM production in the yap1/wwtr1 double mutants is responsible for the reduction in size of epidermal cells. Transplants into 200 blastula stage embryos is easily doable in a single day and the phenotypes of the host yap1/wwtr1double mutant embryos will be evident as the embryos develop. Given the size of the epidermal cell population, virtually every yap1/wwtr1 double mutant would have some wild-type cells in their epidermis. Thus, more than 10 such mosaics embryos could be created in a day and there would be ample control embryos.

4. In the one new experiment presented in the revision (Figure 7), it is not clear why knockdown of yap or taz causes an increase in DNA damage foci, but knockdown of yap/taz together does not increase DNA damage. No explanation is provided for this discrepancy.

---

## [Author Response]

Essential revisions:The most important point seems that there is a lack of direct evidence that the cell size/cell stretching phenotype and the apoptosis phenotype are due to mechanical stress. There needs to be better evidence to support this.Is it known whether different regions of the epidermis experience different levels of mechanical stress? If so, there could be significant differences in mechanical stress among different regions of the epidermis.

This is a very interesting point, and it is currently not well known whether mechanical stress on the epidermis is uniform across the entire embryo.

It is a major challenge to perform robust and reliable in vivo mechanical perturbations to the early embryo. That is why we opted for the perturbations either in the later embryo or utilising cell culture. The latter experiments were generally supportive of our conclusions, but obviously they represent a different system. Therefore, as suggested by the Reviewers, we have toned down the confidence in a number of our conclusions.

Another related idea was to perform transplantation experiments. This is a very powerful approach that has been successfully used in zebrafish. However, in our case, we are analysing a double mutant with 1/16 success rate. Given the trickiness of the transplantation experiment and the low number of successes, this experiment was not feasible given our current time constraints. We have toned down the appropriate conclusions related to this part, reflecting the uncertainty that remains. However, we hope that the Reviewers understand the technical challenges. We emphasise that the significant majority of results presented have not been questioned. Therefore, we believe that the results and conclusions are generally on a solid footing and suitable for publication.

We agree that it is important to evaluate whether differential mechanical stress on the yolk contributes to the observed *yap1;wwtr1* double mutant phenotype (i.e. apoptosis of epidermal cells on the yolk). We therefore elected to test this hypothesis under a more controlled and well-defined environment with cell culture systems. We include new data to test this hypothesis (Figure 7).

The authors need to consider cell area, which they measure, versus cell volume.

The epidermal cells flatten as the embryo progresses through segmentation stages. During the 6-10 somite stages (s.s.), when the apoptosis phenotype appears, the epidermal cells are very thin (about 1 mm). Thus, the measured area approximates cell volume. We have clarified in the text that cell size refers to the apical area (line 155).

How dependent is periderm growth on proliferation of the EBCs at this stage of zebrafish development? The authors should comment this point.

EBCs are highly proliferative at this stage driven by ΔNP63 as shown by Lee & Kimelman (2002). In the absence of ΔNP63, which is expressed in the EBCs but not the peridermal cells, EBC proliferation is reduced, but periderm growth does not appear to be impeded by this perturbation prior to 20 hpf – when apoptosis of the EBCs are observed in *yap1;wwtr1* double mutants.

There was some confusion from the reviewers because you mention in the abstract previous work on the ventral fin while that region was not examined in this study. This is an example of where the broader audience would benefit from some further details/text changes.

We have added introductory texts to give more context as to why our present focus had shifted away from the ventral fin to the epidermis on the yolk (lines 56-61).

Reviewer #1 (Recommendations for the authors):Yap1 and Wwtr1 are important, partially redundant proteins involved regulation of organ size and have been extensively studied. Yap1 and Wwtr1 are known to prevent apoptosis. In the adult mouse epidermis, Yap1/Wwtr1 regulate cell proliferation during wound healing. The role of Yap1/Wwtr1 in epidermis development is unknown and the topic of this study.The manuscript by Lai et al., focuses on the role of Yap1/Wwtr1 in the development of the zebrafish epidermis. The authors present convincing data that epidermal basal cells (EBCs) exhibit reduced cellular growth and increased DNA damage and apoptosis in Yap1/Wwtr1 double mutant zebrafish embryos. The major weakness of the study is the underlying mechanisms of these phenotypes is not clearly established.The data presented in the first five figures is convincing. Yap1/Wwtr1 are necessary for the increase in cell size of EBCs and prevent apoptosis of these cells in the post-gastrulation zebrafish embryo. The authors hypothesize that increasing mechanical stress on the Yap1/Wwtr1 mutant EBC cells causes the apoptosis.

We are pleased that the Reviewer finds the significant majority of our conclusions to be well supported by the presented findings.

In figure six, the authors try a nice in vivo experiment in which they increase mechanical stress in the head of the developing zebrafish embryo by injecting a bolus of mineral oil into the brain. However, this manipulation did not result in the expected increase in apoptosis. It is presumed that this experiment was performed in wild-type embryos as the authors do not specify the genotype. It is not clear why the authors did not try this experiment in Yap1/Wwtr1 mutant embryos as they should be more sensitive to this manipulation according to the author's model.

We thank the Reviewer for this suggested experiment. Unfortunately, the technique has been developed and tested for embryos after 20 hpf, which is beyond the viability of the double mutants. In addition, when the experimental results carried out on WT embryos proved negative, we did not pursue it further as Singapore and India were under strict border controls due to the Δ SARS-COV2 variant – the fish were in Singapore, while the technique was performed in India. We have toned down our conclusions on this experiment to reflect the remaining uncertainty in our results.

In the latter part of figure 6, the authors plate human keratinocyte cells on substrates of different stiffnesses. Here, they observe an increase in DNA damage on cells plated on substrates with a stiffness {greater than or equal to}12.0 kPa. These keratinocytes are also prone to DNA damage via UV irradiation, but it doesn't appear that plating cells on stiffer substrates make them more sensitive to UV induced DNA damage. The relevance of these keratinocyte experiments to understanding the phenotype of Yap1/Wwtr1 mutants is not entirely clear.

As stated above, a challenge we faced was performing robust and reproducible mechanical manipulations in vivo. The use of keratinocytes was to have a cleaner system to explore the role of stiffness in the genomic stability – but obviously at the cost of removing from the in vivo system.

We have included an additional part to the discussion (lines 215-224) that outlines the link between the experiments. We also include an appropriate caveat highlighting the limitation of our approach.

The major weakness of the study is the lack of a mechanistic understanding of the cell growth and apoptosis phenotypes. In the discussion, the authors hypothesize that the defect in cell growth is due to a reduction in the density of the ECM. This is based on the observations that ECM density has been shown to increase cell size in other contexts and the fact that the expression of a number of ECM proteins is reduced in Yap1/Wwtr1 mutants. This hypothesis is easily testable via genetic mosaics as one would expect this phenotype to be cell non-autonomous. Wild-type EBCs in Yap1/Wwtr1 hosts should also exhibit a reduction in cellular growth. Conversely, Yap1/Wwtr1 EBCs in a wild-type host should exhibit restored cell growth.

We thank the Reviewer for highlighting this point, but unfortunately this experiment is extremely challenging. We discuss this point above at the beginning of the letter. We now include a caveat regarding this in the Discussion, to make clear where future work can progress (lines 231-232).

The authors suggest that the increase in the ECM and the corresponding increase in the mechanical stiffness of the embryonic environment causes Yap1/Wwtr1 mutant EBCs to apoptose. In this interpretation, which seems to contradict the first hypothesis, Yap1/Wwtr1 mutants do not produce enough ECM for EBCs to reach their normal size, but Yap1/Wwtr1 mutants do produce enough ECM-dependent stiffness to induce EBC apoptosis. The suggested genetic mosaics experiments could also provide evidence for the author's explanation of the apoptosis phenotype. Might one expect an even higher rate of apoptosis of Yap1/Wwtr1 mutant EBCs when they are developing in a more mechanically robust wild-type host?

We agree with the Reviewer that there is tension in the results from our experiments. It appears that Yap1/Wwtr1 has two important functions here: (1) Drive expression of ECM genes; and (2) promote survival under genomic stress. These processes are intertwined with the hypothesis that under increased mechanical stiffness cells experience genomic stress. The suggested experiment of performing a mosaic experiment should disentangle and address this hypothesis.

As outlined above, unfortunately such a mosaic experiment is very challenging to perform, both due to the early developmental stage and the very low probability of gaining cells of correct genotype. Even if the embryo transplantation experiment worked, it’s likely that the results would be confusing as the readouts may be unclear. For example, our key readout is apoptosis of mutant cells which, if the hypothesis is true, becomes absent during sample collection and imaging. This readout means that either transplantation for that embryo was suboptimal, or that we are only observing after the fact that the apoptosed cells have been cleared away.

This is why we decided to robustly test this hypothesis using human-derived keratinocytes, namely the HaCaT cell line. Obviously, we lose the advantage of the in vivo system. However, we have far greater control on the perturbations, making readout clearer. We found that indeed, both depletion of Yap1/Wwtr1 and increasing mechanical stiffness could induce recruitment of γH2AX. We have introduced suitable caveats within the manuscript to make these points clearer.

One issue with the text is ambiguity about what the authors mean by "stress". As in physics, this could mean force per unit area. Alternatively, it could refer to a challenging physiological condition as the term is frequently used in biology. Since this study involves mechanobiology, the authors would need to clearly define what they mean by stress. This ambiguity makes it difficult to always discern the meaning of the text, and it appears that the different meanings are conflated in the text.

We thank the Reviewer for highlighting this issue – we agree that this term is often used ambiguously. We have tried to clarify our terminology in the manuscript. We have endeavoured to make sure that when referring to the sense with regards to challenging physiological conditions, we always use “genomic stress”. Further, to avoid confusion, we do not use the word “stress” explicitly in the manuscript to refer to the physics meaning of force/area.

Relatedly, it is unclear what the authors mean by "genomic stress". Is the genome under greater mechanical stress in Yap1/Wwtr1 mutants? It would also be helpful to state how the authors think increased substrate stiffness increases genomic stress. Is this relationship purely mechanical or is cell signaling and gene expression involved? What is currently known about the link between the mechanical environment and genomic stress?

The question of the link between mechanical environment and genomic stress is a topic of significant interest. Recent work from the Wickström lab (Nava *et al.* Cell 2020) has revealed that nuclear softening can protect the genome from mechanical perturbations in epithelial tissues. This stands in contrast to experiments on cancer lines, that reveal DNA damage under mechanical stress (Denais *et al.* Science 2016, Raab *et al.* Science 2016). We include these references when introducing the concept of genomic stress (line 76), which helps to clarify the comparison of genomic with mechanical stress.

The study is interesting and rigorously performed. However, the authors need to better substantiate their conclusions to merit publication in eLife. Specifically, the authors need to provide a better understanding of the mechanisms underlying the cell growth and apoptosis phenotypes. The beforementioned mineral oil injections into Yap1/Wwtr1 mutants could provide useful data. However, the genetic mosaic experiments could really be clarifying, and these are standard experiments in zebrafish.

As stated above, the mosaic experiments are especially challenging in this case and the expected readouts (*i.e.* apoptosis of mutant cells) may not necessarily address the hypothesis at hand.

Reviewer #2 (Recommendations for the authors):Lai et al. submit a manuscript investigating the role for Wwtr1/Taz and Yap1 in morphogenesis of the posterior body and epidermal fin fold. Through Zebrafish double mutant embryos, the authors present data suggesting that Wwtr1 and Yap1 cooperate to protect epidermal basal cells from DNA damage during morphogenesis, confirming previously published studies. Not only are both Wwrt1 and Yap1 localised to EBC nuclei, but these cells undergo Tp53 independent apoptosis driven by DNA damage. Recruitment of DNA damage marker γH2AX in mutant embryos is also associated with reduced cell size. Using an in vivo ventricular inflation assay and in vitro cell stretching, the authors suggest that increased tissue stiffness may cause DNA damage that is protected against by Wwrt1 and Yap1.The submitted manuscript aims to investigate Wwrt1 and Yap1 in epidermal morphogenesis and presents data suggesting a surprisingly strong need for both proteins to protect Epidermal Basal Cell DNA during somitogenesis. While the authors confirm no proliferation defect in mutant embryos, proliferation is assayed early in the mutant pathology and may not represent the cell dynamics present in later stages where the authors also find EBCs to be smaller. A conflict arises between these data as increased cell death and no significant increase in proliferation might lead to epithelial rupture. Why mutant EBC are smaller is not directly addressed but the authors suggest that embryonic stiffening might underlie epidermal susceptibility to DNA damage, however, there are insufficient control experiments and discussion to make this conclusion.

We thank the Reviewer for their reading of our paper and their constructive critiques. Below, we outline how we address these concerns.

Further experiments and discussion are required to make some of the claims presented here and to improve impact of the work within this hotly studied pathway.1. Results section is not very well motivated. For example, why 16-18 ss? Not entirely clear whether this stage was analysed by previous publication, why exactly that is so important for the phenotype – why that might be crucial in epidermal development as opposed to other stages.

We agree it is important to clearly outline our particularly choice of timing. We have added additional detail on lines 85-90.

2. The text states no apoptosis was seen in the tail as before, however, the image shown in Figure 1 does appear to have TUNNEL staining in the posterior embryo albeit much less than the rest of the embryo. Is there any way to quantify this phenotype and represent the average phenotypes for the brood?

The quantification for this suggestion is reflected in Figure 1B’. It is worth noting that it is less reliable to quantify TUNEL foci of images derived from the 10x lens. The images shown in Figure 1 are representative of our embryos. The focus of this paper is the epidermis phenotype of yap1/wwtr1 mutants, and so a detailed quantification of the temporal-spatial apoptosis paper is beyond the paper scope.

3. No justification for studying 6 ss embryos for detailed looked at EBCs. Is there a constant level of apoptosis across these stages of development or does it change over time?

As apoptosis accumulates over time, the TUNEL signal was localised to apoptotic fragments (very late into apoptosis) and was thus difficult to assess which epidermal cell type was affected (i.e. TP63+ or TP63- or peridermal cells). We have now included more details in the paragraph (lines 84-101) where we introduce these experiments.

4. Figure 1: Single colour panels would be required to show nuclear TUNNEL staining.

We have now included the single colour channels in Figure 1, Supplementary Figure 1.

5. Figure 2: Throughout text and legends figure 2 is said to show Wwrt1 and Yap1 expression but only immunofluorescence is shown. Quantification of immunofluorescence should be added to the main figure to bolster the imaging data.

We have now included the quantification into the main Figure 2, which was originally in the associated supplementary figure_­_.

6. Figure 2 S1: The pattern of apoptosis seen in the TEAD binding mutant is different from that seen in the complete null. Could these differences indicate a spatial difference in Wwrt1/Yap1 function? This is not addressed and could be improved by quantification/comparison with other mutants.

The phenotypes of the TEAD binding mutant is not identical but similar to the complete null ones. The increased apoptosis is in a similar domain of the embryo and timing of apoptosis onset is like the complete nulls. We do expect some variations between the mutant embryos but the results across all our embryos (7 for the complete null and 6 for the TEAD mutant) are consistent. Therefore, our current data does not suggest that there are spatial differences. More detailed future work may reveal more subtle differences.

7. Why proliferation was assayed in mutants at 3-5 ss is not well justified. Ideally 6 ss and 18 ss would be included to confirm no change in proliferation. This is especially important as not only do mutant cells become smaller, then die more often. How enough cells would be made to compensate for cell loss without rupture of the epithelium leaves these data looking conspicuous and should also be addressed in the text as well.

We chose to assay proliferation at these stages as loss of cells to apoptosis at later stages will reduce the number of cells, which potentially confounds with fewer number of ph3 cells. Although documenting a trend would be interesting, it is beyond the scope of the present investigation.

The question of cell compensation does not arise in this scenario as the periderm remains intact and therefore the integrity of the tissue remains intact.

8. Live Capase-8 probe is not cited in results text.

We include in the Methods the use of the Caspase-8 probe, including where we acquired the probe (line 306-310).

9. The role for Tp53 in the embryo/cell or why it is used to distinguish the extrinsic apoptotic pathway is not explained.

Tp53 plays a role in tumour suppression and drives apoptosis. However, we recognise that upon revisiting this experiment, that it does not aid in distinguishing the apoptotic pathways. Therefore, we now omit this from the revision.

10. Figure 4: No statistical tests on image quantifications.

We have now included this in the revised manuscript.

11. Cell size was calculated as the 2D area of EBCs, however this is totally cell size as these cells could take on another shape, becoming more columnar. In this case cell size could be the same but tissue geometry would be different and put different force upon the nucleus.

This is a valid concern. However, in this case the EBCs are exceptionally thin. Their average depth is estimated to be 1 mm. Therefore, the area is a close proxy of the cell size as the size in the third dimension is small. We have clarified this on line 155.

12. Further to comment 7. How can mutant EBCs be smaller and there be no rupturing of the epithelium? Are other parts of the epithelium proliferating more/less to allow for this constraint?

We thank the Reviewer for highlighting this. It is relevant to point out that the periderm remains intact and is the superficial layer that keeps the embryo from rupturing.

13. It is not clear which EBCs were assayed in ventricular injection experiments. The paper is focussed on the posterior embryo, yet assays whether the head epithelium is susceptible to stretch induced DNA damage. While a potential proxy the strengths and weaknesses of this experiment is not provided – which include the very short time scale of stretching. Further, no data are presented that demonstrate the efficacy of this experiment here.

The ventricular inflation technique has been thoroughly tested and its efficacy has been reported elsewhere (Lewis, N. S., et al., 2020). Indeed, the same scientist (Geetika Chouhan) performed these experiments. Unfortunately, local inflation at the yolk area has not been thoroughly tested for its efficacy and robust readouts, thus we did not attempt it.

14. Why gels of these stiffnesses were used has not been justified, nor has the susceptibility of these nuclei to damage at 12Kpa increased compared to greater stiffness been well discussed. How do the collagens within the gels compare to the ECM that EBCs are in contact with? How does the stiffness of the posterior embryo compare to these gels? etc.

As our aim was to address the question on whether stiffer environment could induce genomic stress we sampled a range of hydrogel stiffness for this experiment. While measurements of relative stiffness changes are possible in vivo (e.g. using laser ablation or aspiration) it is challenging to gain absolute numbers on the embryo stiffness (Petridou and Heisenberg, EMBO Journal 2019). Cantilevers have been used to estimate the mechanical properties of the early zebrafish embryo. The measured Young’s modulus is similar to epithelial cells, though this measurements are only approximate (Tomizawa *et al.* Sensors 2019). It’s likely that the embryo falls within the range of stiffness that we measured in cell culture. We have now justified our approach more clearly (lines 215-224).

15. The claims of the paper would need to be toned down throughout the manuscript without further controls being added.

Given the concerns raised by the Reviewers and that some suggested experiments are not currently doable, we agree with this comment. We have adjusted our conclusions to more closely reflect the confidence we have in our experiments (*e.g.* lines 220-222, 231-232, 241-243).

Reviewer #3 (Recommendations for the authors):This important study identifies a new type of anti-apoptotic function for YAP/TAZ proteins in zebrafish. While YAP/TAZ proteins are known to protect cells from apoptosis in many different animal species, the authors identify a novel and specific requirement for YAP/TAZ in preventing apoptosis caused by mechanical forces that flatten the cell nucleus and lead to DNA damage (an inducer of apoptosis). Double yap1, taz/wwtr1 knockout zebrafish embryos reveal an essential function of YAP/TAZ in maintaining cell survival in the basal layer cells of the epidermis, cells that experience significant mechanical stretch during embryonic morphogenesis. It is proposed that mechanical stress/strain in the epidermis leads to genomic stress, which then requires nuclear translocation of YAP/TAZ to ensure cell survival. In addition, YAP/TAZ are shown to promote elastic expansion and flattening of epidermal cells in response to stretch. The findings are an important addition to previous work in mice, where YAP/TAZ have a mainly pro-proliferative role in basal layer skin epidermal cells during development. Thus, in different contexts, mechanical stress/strain can induce nuclear YAP/TAZ to drive either increased cell proliferation (as in mouse skin), or cell expansion/flattening and survival (as in fish embryonic skin), to govern how the tissue responds to the mechanical stimulus.The conclusions are clearly supported by the results.The major strengths are the quality of the zebrafish genetics and phenotypic analysis, as well as the novel insights produced, particularly that the response of zebrafish skin to mechanical stretch is different to that of mouse skin, yet still governed by YAP/TAZ. That mechanical stress leads to genome stress is also a novel finding, and the role of YAP/TAZ in ensuring cell survival in this particular context is also a novel function for these proteins.These findings will be of major significance to both developmental biology and cancer.

We thank the Reviewer for their overall positive review of our manuscript.

[Editors' note: further revisions were suggested prior to acceptance, as described below.]

The manuscript has been improved but there are some remaining issues that need to be addressed, as outlined below:1. The major deficiency of the original submission of this manuscript was a lack of direct evidence that the cell size/cell stretching phenotype and the apoptosis phenotype in yap1/wwtr1 double mutant zebrafish are due to mechanical stress. The authors have addressed this issue not with new data, but by making their conclusions more circumspect. The problem is that the bold conclusions that made the study more broadly interesting are not adequately supported by the data. For example, the impact statement for the manuscript, "Substrate rigidity modulates genomic stress in the epidermal basal cells of the developing zebrafish embryo and Yap1 and Wwtr1 are required for its survival" is not clearly demonstrated by the data in the manuscript. Thus, the study in its current form is of less general interest and may be better suited for a different journal.

We thank the reviewers for this feedback. We have revised the concluding statements to capture the results of the experiments presented.

“Thus, our experiments suggest that substrate rigidity modulates genomic stress in epidermal cells, and that Yap1 and Wwtr1 promotes their survival.”

As support for the relationship between substrate rigidity and genomic stress is limited to culturing HaCaT cells on different substrate rigidity, we have corrected the first part of the conclusion by not alluding this relationship to the developing epidermis, which should rightly be tested with mosaic experiments as suggested below.

The second part of the conclusion reflects the observed phenotype of the double *yap1;wwtr1* mutants.

Along with the above changes, a concern is raised about the generality of our results. Our findings that Yap1/Wwtr1 play an important role in cell survival in cells under mechanical stress is a very important one. It has clear potential impact, from developmental biology to cancer research.

Therefore, even though our conclusions are toned down, they will still be of broad interest.

2. The author's response is that further experiments are not really feasible. However, this is not entirely true. Genetic mosaic experiments are routine in zebrafish. It is true that transplantation of yap1/wwtr1 double mutant cells into wild-type embryos is challenging because only 1/16 donor embryos will be double mutant, and one doesn't know the genotype of the embryos at the time of transplantation. There are ways to perform the experiment by allowing the donor embryos to develop along with the associated mosaic embryos created by the transplantation. The double mutants could be identified as the phenotype becomes apparent and the associated mosaic embryos could be characterized in detail. Thus, this experiment is difficult, but not impossible.

We agree with the reviewer that this experiment is difficult, but not impossible. However, we face two hurdles at the moment.

First, despite a number of trials, the relevant fish in our system are not spawning well. We likely need to out-cross the fish to return them to laying well, which will take > 3 months.

A further issue here is that while this experiment is doable in a technical sense, it requires significant hands-on work from dedicated and trained lab personnel. Due to Covid, much of the work on this project has been significantly delayed and key lab personnel have subsequently moved onto new positions. Therefore, getting the expertise together is a major challenge at present.

Given the above issues, it is not possible to do the requested experiments within a reasonable timescale. We have therefore toned down our conclusion on this part (see [1]). Despite this, we stress that the manuscript still has numerous strong experiments that provide support for our key conclusions regarding Yap/Wwtr1 and cell survival under mechanical stress. Hence, we hope that the reviewers can accept the manuscript in this current version given the constraints we face.

3. The converse experiment of transplanting wild-type cells into the double mutants is quite doable. Such an experiment could be done to test whether the reduction in ECM production in the yap1/wwtr1 double mutants is responsible for the reduction in size of epidermal cells. Transplants into 200 blastula stage embryos is easily doable in a single day and the phenotypes of the host yap1/wwtr1double mutant embryos will be evident as the embryos develop. Given the size of the epidermal cell population, virtually every yap1/wwtr1 double mutant would have some wild-type cells in their epidermis. Thus, more than 10 such mosaics embryos could be created in a day and there would be ample control embryos.

This experiment is facing similar challenges to point [2] above.

4. In the one new experiment presented in the revision (Figure 7), it is not clear why knockdown of yap or taz causes an increase in DNA damage foci, but knockdown of yap/taz together does not increase DNA damage. No explanation is provided for this discrepancy.

We thank the reviewer for pointing out this omission. We have added this explanation in lines 197-204, reproduced here:

“We found that double knockdown of *YAP1* and *WWTR1* did not deplete their protein expression, unlike the single *YAP1* or *WWTR1* knockdowns (Figure 7A). Accordingly, the depletion of YAP1 or WWTR1 resulted in greater expression of γH2AX by Western Blot (Figure 7A), and more γH2AX foci (Figure 7B). However, we could not test the redundancy of YAP1 and WWTR1 as the double knockdown was incomplete, which may explain the unchanged γH2AX levels in the double knockdown compared to control. This observation suggests a role for YAP1/WWTR1 in protecting the genome from DNA damage.”